# Anticoagulation Monitoring with Activated Partial ThromboPlastin Time and Anti-Xa Activity in Intensive Care Unit Patients: Interest of Thrombin Generation Assay

**DOI:** 10.3390/ijms231911219

**Published:** 2022-09-23

**Authors:** Paul Billoir, Thomas Elie, Jerrold H. Levy, Emmanuel Besnier, Bertrand Dureuil, Benoit Veber, Véronique Le Cam-Duchez, Thomas Clavier

**Affiliations:** 1Normandie Univ, UNIROUEN, INSERM U1096, Rouen University Hospital, Vascular Hemostasis Unit, F 76000 Rouen, France; 2Department of Anesthesiology and Critical Care, Rouen University Hospital, F 76000 Rouen, France; 3Departments of Anesthesiology, Critical Care, and Surgery (Cardiothoracic), Duke University School of Medicine, Durham, NC 27710, USA; 4Department of Anesthesiology and Critical Care, Rouen University Hospital, Normandie Univ, UNIROUEN, INSERM U1096, F 76000 Rouen, France

**Keywords:** coagulation tests, inflammation, intensive care unit, heparin, activated partial thromboplastin time, factor Xa, thrombin generation

## Abstract

Current guidelines recommend monitoring the anticoagulant effect of unfractionated heparin (UFH) by measuring anti-Xa activity rather than activated partial thromboplastin time (aPTT) in intensive care unit (ICU) patients. The primary objective of this study was to evaluate the correlation of aPTT, anti-Xa activity, and thrombin generation in UFH-treated ICU patients. A prospective observational pilot study was conducted in adult surgical ICU patients treated with UFH. aPTT and anti-Xa activity were monitored daily. The therapeutic target was aPTT between 50 s and 84 s, and/or anti-Xa between 0.3 and 0.7 U/mL. Correlation among aPTT, anti-Xa activity, and thrombin generation was determined by measuring endogenous thrombin potential (ETP), with the inflammatory response evaluated. C-reactive protein (CRP) was used as a marker of inflammatory response. The plasma of 107 samples from 30 ICU patients was analyzed. The correlation between aPTT and anti-Xa activity was 0.66, CI_95%_ [0.54;0.76] (*p* < 0.0001). Although thrombin generation, aPTT, and anti-Xa were correlated with inflammatory responses, the correlation was higher with thrombin generation and anti-Xa activity compared to aPTT. When aPTT was in a therapeutic range, a low thrombin generation was observed but was 50% inhibited when anti-Xa was in a therapeutic range. Coagulation testing with aPTT, anti-Xa correlated with thrombin generation. A 50% decrease in thrombin generation was observed when anti-Xa was within a therapeutic range. Further work is needed to evaluate coagulation biomarker responses and clinical outcomes in specific ICU populations.

## 1. Introduction

Multiple pharmacokinetic and biological factors influence anticoagulation requirements in critically ill intensive care unit (ICU) patients. Further, several methods are used to monitor anticoagulation with unfractionated heparin (UFH). Anti-Xa activity reflects the concentration of heparin in the blood, while activated partial thromboplastin time (aPTT) is a direct indicator of the biological efficacy of UFH by measuring increases in clotting time. International guidelines for monitoring anticoagulation by UFH in the specific population of critically ill surgical patients are not available. In France, current recommendations suggest that UFH anticoagulation should be monitored by anti-Xa activity rather than aPTT in ICU patients [1], recommendations that are based on single-center and often retrospective studies of small subject numbers that did not include ICU patients [2,3]. One of these studies concluded, “*It remains unclear whether aPTT, anti-Xa, or endogenous thrombin potential provides the most accurate measurement of the global coagulation status of patients receiving UFH*” [2]. Therefore, the level of evidence to preferentially monitor anti-Xa activity in ICU patients who often have a significant inflammatory response is low.

Several studies have highlighted the potential role of fibrinogen and factor VIII in influencing aPTT measurements. aPTT is shortened in the presence of elevated factor VIII [4] or elevated fibrinogen [5]. Anti-Xa measurement is a direct UFH activity, with indirect quantification of antithrombin activity. Previous study have reported conflicting results for aPTT and anti-Xa measurement of heparin [6]. Several recommendations advocates if heparin resistance is suspected, anti–factor Xa can be used to measure the heparin level [7]. The impact of inflammation on the discordance between aPTT and anti-Xa activity in ICU patients has not been described in ICU patients [5,6,8]. The thrombin generation assay (TGA) evaluates the activation and inhibition of the clotting system [9,10,11,12] and represents a direct test of hemostatic effect reported for low molecular weight heparin and UFH monitoring [13]. Recent data suggest that a 50% reduction in endogenous thrombin potential (ETP) represents a target level of heparin anticoagulation [14]. The primary objective of this pilot study was to evaluate the correlation between aPTT and anti-Xa activity, and the anticoagulation response with thrombin generation assay in patients treated with UFH. The secondary objective was to evaluate the inflammatory response in ICU patients with impact in thrombin generation testing in this specific population, and its correlation to standard coagulation testing.

## 2. Results

### 2.1. Patients’ Characteristics

A total of 107 blood samples were obtained from 32 patients. The number of blood samples was between 2 and 4 for each patient, and the blood samples were collected between 1 and 4 days of hospitalization. The characteristics of the population are summarized in Table 1. Two patients requiring a prophylactic anticoagulation target were excluded. Therapeutic anticoagulation treatment included the following patient issues: 17 abdominal aortic aneurysm surgeries, 5 atrial fibrillation, 5 thromboembolic events, and 3 other reasons. The reasons for ICU admission included 7 urgent surgeries, 4 post-surgery monitoring, 2 traumatic injuries, 2 cardiogenic shock, and 2 other reasons. Mortality in our ICU patients was 12.5% (4 of 32). Of the 30 healthy volunteers, 14 were male, with a median age of 38.2 ± 11.8 years.

### 2.2. aPTT, Anti-Xa Activity, and Their Correlation with Inflammatory Response

In the spiked pool of plasma from healthy volunteers, we observed a correlation between aPTT and anti-Xa activity with increasing doses of UFH (r = 0.99, CI_95%_ [0.93; 1.00]; *p* = 0.0004; cf. Figure 1A). In ICU patients we also found a correlation between aPTT and anti-Xa activity (r = 0.66, CI_95%_ [0.54; 0.76]; *p* < 0.0001; cf. Figure 1B). As expected, CRP was correlated with fibrinogen (r = 0.51, CI_95%_ [0.36; 0.64]; *p* < 0.0001) and FVIII (r = 0.41, CI_95%_ [0.23; 0.56]; *p* < 0.0001).

Concerning hemorrhagic or thrombotic complications, six samples were excluded because the presence or absence of complications was not correctly identified at the time of sampling (analysis on 101 samples). aPTT and anti-Xa were concordant in 49 samples (48.5%), and their levels did not appear to be predictive of a hemorrhagic or thrombotic complication at the time of sampling (cf. Table 2). An increased CRP was observed in aPTT < T/anti-Xa < T (173 mg/L [104–255]) and aPTT = T/anti-Xa < T (143 mg/L [80–190]) compared to aPTT > T/anti-Xa = T (52 mg/L [21–77]) (Appendix A). No difference was observed in the inflammatory biomarker in the anticoagulated group.

### 2.3. Thrombin Generation Assay

In spiked plasma, thrombin generation was decreased with increased anticoagulant target (Appendix A). When anti-Xa was in the therapeutic range, in the presence of 5 pM of TF, Thrombin generation was not observed. The results of thrombin generation results are shown in Table 3. Briefly, a significant difference was observed between the control group and ICU patients, in lagtime and in TTP at 5 pM of TF. Moreover, we observed a decreased thrombin generation in Lagtime, TTP, ETP, peak, and velocity at 20 pM of TF. When we compared different results of aPTT and Anti-Xa, ICU underdosed patients had only increased lagtime and TTP. At 20 pM of TF, patients with therapeutic anticoagulation or higher levels, in aPTT, had significant decreased compared to control and underdosed aPTT/anti-Xa patient. An anti-Xa under therapeutical range was associated with a global incoagulable plasma (Figure 2). In patients’ plasma, we observed significant correlations between thrombin generation parameters, aPTTratio, and anti-Xa activity, at both concentrations of tissue factor studied (Appendix A). No correlations were observed between lagtime and aPTT at 5pM of TF. Moreover, no correlation were observed between lagtime, aPTT, and anti-Xa at 20 pM of TF. The association with thrombin generation parameters was higher for anti-Xa than aPTT. However, no correlation was observed between thrombin generation and FVIII. For each case of hemorrhagic or thrombotic complication (cf. Table 2), no particular TGA profiles (increase or decrease in TGA parameters) were observed. When we compared these results with a 50% ETP reduction, we observed ETP under target with anti-Xa under a normal range, and a significant difference when aPTT was under target (Appendix A).

## 3. Discussion

In our current study, aPTT and anti-Xa activity correlated with the intensity of thrombin generation, with a greater correlation with anti-Xa activity than aPTT in ICU patients. CRP, an acute-phase inflammatory protein, increases in nonseptic patients. CRP did not differentiate septic complications in patients with trauma, but levels remained elevated in the late post-traumatic period [15].

Of note is the potential for variable responses to UFH in ICU patients. Furthermore, heparin resistance is not well defined [16], potentially due to multiple causes, including decreased antithrombin levels, especially in ICU patients [7], and the aPTT response to UFH is not linear at higher heparin concentrations [17]. A report comparing aPTT and anti-Xa activity in an ICU population found a concordance between both parameters of only 63% of samples compared to 48.5% in our study [18]. However, this study did not evaluate the impact of inflammatory responses on the discordance between these parameters. In some clinical scenarios with a constitutional increase in aPTT (e.g., cirrhosis and antiphospholipid syndrome) it seems logical to preferentially use anti-Xa activity for UFH monitoring. However, for patients with normal aPTT preoperatively or on ICU admission, there are, to our knowledge, no studies investigating and comparing the optimal level of both these parameters in the subgroup of ICU patients with systemic inflammation, including trauma, surgery, and/or sepsis. Our findings suggest that neither aPTT level nor anti-Xa activity was predictive of bleeding or thromboembolic complications, although the study was not designed to identify complications. We focused on surgical ICU patients who are at higher risk of bleeding complications due to their recent surgery than medical ICU patients, regardless of the level of anticoagulation.

The thrombin generation assay (TGA) is an increasingly used laboratory test that provides a reliable hemostatic profile and a high association with bleeding phenotype [19,20,21]. A previous study demonstrated a global decrease in thrombin generation in UFH [22]. There is a significant correlation between the intensity of anticoagulation and ETP in patients treated with UFH. However, there was a poor correlation between ETP and either aPTT or anti-Xa levels, there was a higher correlation between UFH dose and the degree of ETP suppression with UFH treatment [2]. With a global functional hemostasis evaluation, we first observed a reduced thrombin generation in spiked plasma compared to ICU patients. The contribution of a thromboinflammatory response could explain the increased thrombin generation in ICU patients. Thrombin generation was completely inhibited in samples whose anti-Xa levels were therapeutic but not in samples whose aPTT were in therapeutic ranges. Decreases in thrombin generation are associated with increased bleeding, potentially suggesting that anti-Xa activity at the usual therapeutic level could lead to an increased risk of bleeding in ICU patients [23]. Currently, there is no recommendation to use aPTT or anti-Xa to monitor UFH treatment in critically ill patients despite recommendations for non-critically ill hospitalized patients [24]. In view of the large interindividual differences in heparin responsiveness, heparin dosage should be individualized, as noted by Hemker et al. who recommended using an ETP of approximately 50% of the average of normal population values [14], i.e., 600–700 nM min [25]. We observed an ETP below target with anti-Xa activity for adequate levels but not with aPTT. However, another report of a recent retrospective study described no correlation between anti-Xa or aPTT values with serious bleeding or thrombotic complications [26]. Furthermore, there is no work specifically addressing these levels in ICU patients. This may partly explain the discordance between high-risk bleeding profile on TGA and normal anti-Xa activity in our study [22,26,27].

Our work has several limitations. First, the evaluation period is relatively short with a small number of patients. However, we were able to examine the impact of UFH in TGA as a proof-of-concept study. Second, although we used TGA to evaluate anticoagulation intensity with inflammatory responses, only one study has reported the clinical impact of high FVIII level (mimicking an acute phase response) on heparin monitoring with TGA [28]. Although the authors reported decreased thrombin generation with increased FVIII levels, this study was in vitro performed with FVIII and UFH spiked plasma and not from treated patients [29]. Their ex vivo method was not validated to determine the presence of anticoagulants and the relevance of using TGA to monitor UFH remains controversial [30]. A previous study demonstrated a significant impact of thrombin generation with heparin anti-Xa levels >0.5 U/mL. However, in our study, the objective of TGA use was to obtain a global assessment of coagulation according to the levels of aPTT and anti-Xa activity and not to accurately monitor anticoagulation. A possible explanation is FVIII increases in ICU patients enhances the speed of thrombin formation without enhancing global coagulation [31].

## 4. Materials and Methods

### 4.1. Population Selection

We conducted a prospective single-center observational pilot study in the surgical ICU of a French University Hospital from 5 June to 15 August 2016. All adult patients hospitalized in the surgical ICU or post-operative ICU and who received therapeutic anticoagulation with intravenous UFH for more than 12 h were included. Exclusion criteria were age under 18 y/o, dialysis with UFH anticoagulation, history of heparin allergy, and thrombophilia or constitutive hemorrhagic disease. The study protocol was approved by our University Hospital’s non-interventional research ethics committee (n° E2016-38). An informed consent was obtained from patient.

### 4.2. Study Procedures

In all patients, aPTT, anti-Xa activity, fibrinogen, factor VIII (FVIII), and C-reactive protein (CRP) were measured each morning. Thrombin generation was also measured (cf. *infra*). aPTT was used to monitor heparin therapy and adapt UFH dosages (usual department practice). The curative anticoagulation target was 1.5–2.5 for aPTT ratio (aPTT_patient_/aPTT_control_ with aPTT control = 33.5 s.) [32] and 0.30–0.7 UI/mL for anti-Xa activity [33].

We defined 3 groups of anticoagulation, according to the aPTT results:
aPTT < 50.3 s. (aPTT < T): underdosed UFH target;50.3s. ≤ aPTT ≤ 83.8 s. (aPTT = T): under therapeutical target;aPTT > 83.8 s. (aPTT > T): overdosed UFH target.

We defined 3 group of anticoagulation, according to the anti-Xa results:
anti-Xa < 0.3 UI/mL (anti-Xa < T): underdosed UFH target;0.3 UI/mL ≤ anti-Xa ≤ 0.7 UI/mL (anti-Xa = T): under therapeutical target;Anti-Xa > 0.7 UI/mL (anti-Xa > T): overdosed UFH target.

After each sampling time, we checked whether any hemorrhagic or thrombotic complications had been diagnosed within the previous 24 h. Monitoring was stopped in the event of heparin induced thrombocytopenia, discontinuation of anticoagulation, and discontinuation of daily sampling. We also the effects of a 50% ETP reduction, as proposed by Hemker et al. [14].

To evaluate, in vitro, in normal conditions, a pool of normal plasma from 30 healthy volunteers was spiked with different doses of UFH (Heparin Choay, 5000 UI/mL) to obtain several target ranges. Healthy volunteers had no history of bleeding and thrombosis. Then, aPTT, anti-Xa activity, and TGA were performed for each dose of UFH.

### 4.3. Blood Collection

Plasma samples were collected after consultation. Platelet poor plasma (PPP) samples for TGA were obtained from blood samples, taken by antecubital venipuncture, and collected in vacutainer tubes containing buffered 0.109 M trisodium citrate (Greiner) (1 part of citrate 3.2%/nine parts of blood) with a needle of 21 gauge. Two citrated tubes were collected. PPP was prepared 1 h after sample by double centrifugation of citrated blood for 15 min at 2250 g. Storage of PPP in aliquots was at −80 °C until analysis, as recommended by the “Groupe Français d’études sur l’Hémostase et la Thrombose”, to perform aPTT, anti-Xa and TGA measurement (www.geht.org accessed on 1 May 2016).

### 4.4. Laboratory Assays

Testing of aPTT, fibrinogen, and anti-Xa activity was performed on STAR Max automate (Diagnostica Stago-Asnières-France) with, respectively, PTT-Automate, liquid Fib, and liquid anti-Xa reactants, without exogenous antithrombin (Diagnostica Stago, Asnières, France). FVIII was performed on STAR EES (Diagnostica Stago-Asnières-France) with Standard Human Plasma and Pathromtin SL (Siemens, Marburg, Germany). CRP was assayed with turbidimetry on COBAS 8000 (Roche Diagnostic, Meylan, France).

Thrombin generation was evaluated by phospholipid (4 µmol/L) and different concentrations of tissue factor (TF) with 5 or 20 pmol/L, platelet-poor plasma (PPP) reagent or PPP reagent high, respectively (Diagnostica Stago, Asnières, France) [34]. Thrombin generation was measured by Calibrated Automated Thrombography and Fluorocan Ascent Fluorometer (Thermoscientific Labsystems, Helsinki, Finland). The following parameters of thrombin generation curve were considered: lagtime (first trace of thrombin formation), time to peak (TTP; time necessary for thrombin maximal value), thrombin peak (maximal thrombin concentration), ETP (endogenous thrombin potential: area under the thrombin time concentration curve), and velocity (thrombin speed formation, also calculated: peak/(TTP-lagtime)). We used plasma from healthy volunteers for the control group. A reduction of 50% ETP corresponded to 672 nM.min and 831 nM.min (with 5 pM and 20 pM of tissue factor, respectively).

### 4.5. Statistical Analysis

The values are presented as absolute and percentage values (n (%)) for qualitative variables and as mean ± standard deviations for quantitative variables. Correlations were studied with the Pearson test. Quantitative variables were compared with *t*-test and Kruskal–Wallis test with post hoc Dunn’s test. Qualitative variable was compared with chi square. The significance threshold was set at 0.05, and all statistics were produced using GraphPad PRISM software (San Diego, CA, USA (v 8.0)).

## 5. Conclusions

In summary, with limited data on therapeutic UFH monitoring in ICU patients, anti-Xa had a greater impact on inhibiting thrombin generation than aPTT in our study, although aPTT may better reflect the hypercoagulability of acute inflammation. Additional prospective studies are needed, including thrombin generation in ICU patients to objectively compare the clinical relevance of anticoagulation monitoring.

## Figures and Tables

**Figure 1 ijms-23-11219-f001:**
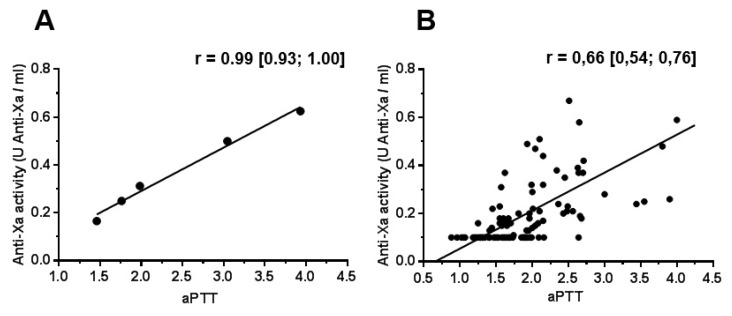
**Correlation between aPTT and anti-Xa activity in a spiked normal plasma pool (A) and in ICU patients (B)**. Data are presented as Pearson coefficient with 95% confidence interval. aPTT: activated partial thromboplastin time.

**Figure 2 ijms-23-11219-f002:**
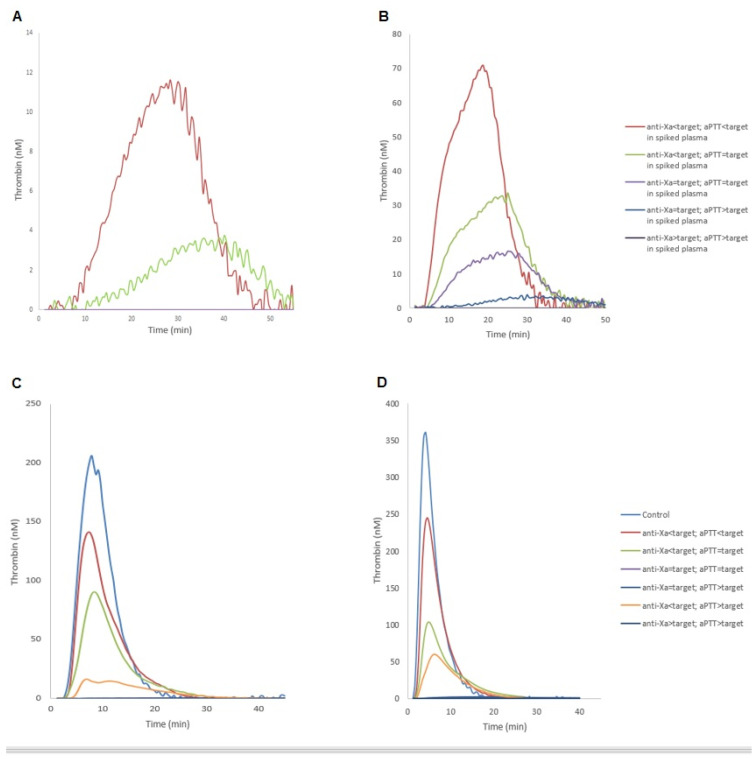
Thrombin generation curve according to the levels of anti-Xa activity and aPTT in spiked heparin plasma (**A**,**B**) and in ICU patients (**C**,**D**). Corresponding to 5 pM (**A**,**C**) and 20 pM (**B**,**D**) of tissue factor to start reaction. The control group of ICU patients consisted of pooled plasmas without anticoagulant treatment. aPTT: aPTT: activated partial thromboplastin time; ICU, intensive care unit; TF, tissue factor.

**Table 1 ijms-23-11219-t001:** **Characteristics of the population**. Data are presented as absolute (%) or mean (± SD) values. ICU: intensive care unit. SAPS II: simplified acute physiology score II. aPTT: activated partial thrombin time. CRP: C reactive protein. DVT: Deep Vein Thrombosis. PE: Pulmonary Embolism.

Number of Patients	30
Age (years)	62.7 ± 15.8
Male	24 (80%)
Body mass index (kg/m^2^)	27.4 ± 6.7
SAPS II score	34.3 ± 14.5
**Reason for ICU admission**	
Post-operative ICU	13 (43.4%)
Unprogrammed ICU admission	17 (56.6%)
**Indication of anticoagulation**	
Vascular surgery requiring anticoagulation	17 (56.6%)
Transitory atrial fibrillation	5 (16.6%)
DVT/PE development	4 (13.3%)
Arterial thrombosis	1 (3.3%)
Others	3 (10%)
**Duration of stay (days)**	
In ICU	12.8 ± 19.6
In hospital	28.1 ± 30.0
**Survival 28 days after ICU admission**	29 (96%)
aPTT (s.) (N: 28.5–38.5)	57.0 [48.6–72.0]
Anti-Xa (UI/mL) (N: 0.35–0.7UI/mL)	0.13 [0.1–1.6]
Fibrinogen (g/L) (N: 2–4 g/l)	6.0 [5.0–7.3]
FVIII (%) (N: 50–150%)	235 [183–368]
CRP (mg/L) (N < 5 mg/L)	127.5 [57–188.3]

**Table 2 ijms-23-11219-t002:** **Bleeding and thromboembolic complications according to anticoagulation target.** Results are presented as absolute values; aPTT, activated partial thromboplastin time ratio.

	aPTT < 50.3 s.	50.3 s.< aPTT < 83.8 s.	aPTT > 83.8 s.
**Anti-Xa activity** **<0.3 UI/mL**	n = 32 2 bleeding complications 1 thromboembolic complication	n = 29 2 thromboembolic complications	n = 14 1 bleeding complication
**0.3 UI/Ml <** **Anti-Xa activity** **<0.7 UI/mL**	n = 2 No complication	n = 8 No complication	n = 6 No complication
**Anti-Xa activity** **>0.7 UI/mL**	n = 0	n = 1 No complication	n = 9 1 thromboembolic complication

**Table 3 ijms-23-11219-t003:** **Thrombin generation in anticoagulated patients.** TGA condition: Tissue factor concentration used to start thrombin generation. aPTT: activated partial thrombin time. ETP: Endogenous thrombin potential. IC: incoagulable. ICU: Intensive Care Unit. N: number of patients in each group. T: target. aPTT target was between 50.3 s and 83.8 s. Anti-Xa target was between 0.3 and 0.7 UI/mL. TGA: Thrombin Generation Assay. UFH: Unfractioned Heparin. ^a^ significant difference with control group. ^b^ significant difference with aPTT < T and Anti-Xa < T. TTP: Time to peak.

Group	UFH Target	TGA Condition	Lagtime (min)	TTP (min)	ETP (nM.min)	Peak (nM)	Velocity (nM/min)
Control	NA	5 pM	3.13 [2.92–3.33]	6.67 [5.94–7.82]	1344 [1190–1575]	199 [157–253]	52.6 [37.3–86.0]
	NA	20 pM	1.67 [1.67–2.08]	4.17 [4.07–4.7]	1662 [1446–1782]	332 [283–398]	138.7 [108.9–174.7]
ICU	NA	5 pM	5.84 [4.58–9.07] ^a^	9.58 [7.29–15.0] ^a^	1199 [553–1520]	200 [41.5–290]	64.25 [8.18–124.2]
(N = 99)	NA	20 pM	2.92 [2.14–3.75] ^a^	5.83 [4.58–8.75] ^a^	1264 [465–1557] ^a^	231 [52–304] ^a^	84.2 [11.5–140] ^a^
ICU	aPTT < T	5 pM	5.42 [4.17–8.96] ^a^	8.12 [6.67–13.12] ^a^	1360 [826–1489]	251 [78–297]	98.4 [124.6]
(N = 27)	Anti-Xa < T	20 pM	2.09 [2.08–2.92] ^a^	4.79 [4.17–5.62]	1442 [1227–1553]	291 [235–338]	128 [89.2–161]
ICU	aPTT = T	5 pM	5.94 [5.0–8.7] ^a^	9.69 [8.07–16.3] ^a^	1170 [451–1649]	144 [25–298]	45.2 [6.0–119.2]
(N = 47)	Anti-Xa < T	20 pM	2.72 [2.5–3.75] ^a^	6.25 [4.59–8.75] ^a,b^	1154 [450–1617] ^a^	210 [51–304] ^a^	53.4 [7.5–122.4] ^a,b^
ICU	aPTT = T	5 pM	IC	IC	IC	IC	IC
(N = 9)	Anti-Xa = T	20 pM	IC	IC	IC	IC	IC
ICU	aPTT > T	5 pM	9.58 [8.12–12.08] ^a^	19.37 [11.87–44.37] ^a^	446 [316–577]	25 [1–72]	11.3 [3.4–19.2]
(N=4)	Anti-Xa < T	20 pM	3.44 [2.97–3.91] ^a^	8.65 [6.77–14.74] ^a,b^	595 [171–1182] ^a^	71 [9–139] ^a,b^	14.1 [1.26–49.7] ^a,b^
ICU	aPTT > T	5 pM	IC	IC	IC	IC	IC
(N = 12)	Anti-Xa = T	20 pM	IC	IC	IC	IC	IC

## Data Availability

Not applicable.

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
