# Peer review of "Anticoagulation Monitoring with Activated Partial ThromboPlastin Time and Anti-Xa Activity in Intensive Care Unit Patients: Interest of Thrombin Generation Assay"

_ijms, 2022, doi:10.3390/ijms231911219_

Round 1

Reviewer 1 Report

Comments to the Authors:

The manuscript by Dr. Billoir on “Anticoagulation monitoring with activated partial thromboplastin time and anti-Xa activity in intensive care unit patients: interest of thrombin generation assay” describes s to evaluate the correlation of activated partial thromboplastin time (aPTT), anti-Xa activity, and thrombin generation in UFH treated ICU patients. The article can be modified a little by incorporating few suggestions.

The study is well planned and designed to reach the aim, and the discussion goes well with the results. Authors need to incorporate minor suggestions to improve the manuscript by including some details about introducing anti-Xa.

Minor concerns:

• Authors need to explain anti-Xa in the introduction to clarify its role and importance in intensive care unit patients.

• Please undergo a thorough check of the manuscript for typographical and grammatical errors.

Author Response

Dear Editor,

First and foremost, on behalf of all authors I would like to thank you and the Reviewers for the valuable comments and suggestions, which have significantly contributed to improving the overall quality of the manuscript. Each comment has been addressed below in point-by-point responses.

Best regards,

Dr Paul Billoir

  • Authors need to explain anti-Xa in the introduction to clarify its role and importance in intensive care unit patients.

We understand reviewer suggestions. We have added in manuscript :

« aPTT is shortened in the presence of elevated factor VIII [4] or elevated fibrinogen [5]. Anti-Xa measurement is a direct UFH activity, with indirect quantification of antithrombin activity. Previous study have reported conflicting results for aPTT and anti-Xa measurement of heparin[6]. Several recommendations advocates if heparin resistance is suspected, anti–factor Xa can be used to measure the heparin level [7].”

  • Please undergo a thorough check of the manuscript for typographical and grammatical errors.

We thank the reviewer for this helpful comment. The manuscript has been thoroughly edited throughout for spelling, grammatical accuracy and overall clarity of expression by our native English speaking Medical Editor at Rouen University Hospital. Moreover, one author is a native English speaker. We sincerely hope that the revised manuscript now achieves the level of English required.

Reviewer 2 Report

This manuscript examines the anticoagulation monitoring with aPTT and Anti-Xa activity in ICU patients, with an emphasis on Thrombin Generation Assay. The manuscript is written in standard language and flow.  Standard reagents and tools were used for various experiments. This article has value and importance in the field that should be considered positively.  Some of the minor issues are as follows:

INTRODUCTION: It would be useful to write a paragraph on the UFH, aPTT, anti-Xa activity, and Thrombin Generation Assay in the particular population of patients addressed in this study. At present, the introduction describes general “ICU cases”; however, the authors have used a selected set of patients. Therefore, it would be more meaningful to address the monitoring principles in this selected set of patients.

RESULTS:

1).  Line 70. It seems multiple samples were taken from each patients, since 107 samples were obtained from 32 patients. Therefore, it would be useful to mention how many samples (minimum-maximum) were taken and at what time point they were taken and the rationale for this approach.

2). Line 78. More information is needed for the control group. Describe the demographics of the healthy volunteers; whether they match with the corresponding parameters of the ICU cases.

3). Line 86. It is mentioned that 49 samples were concordant; but how many patients does it correspond to? 

In general, it is confusing to use the number of samples, instead of the number of patients in the results and analysis. Since more than one sample was likely taken from each patient, the data may be skewed; the authors should consider including the number of patients within parenthesis, whenever the sample numbers are mentioned.

METHODS:

1). Line 229. Mention the method of spiking. What doses of UFH were used for spiking and how was the spiking done (volume and conditions).

2). Line 233. Explain how much blood per draw was collected.

3). Line 238. Mention if storing PPP in -80 oC would affect the analysis of biomarkers.

4). Line 272-274. Remove the Section 6. Patents

5). Line 287. Mention who received the funding, the name of the funder, and the funding/project number.

It is confusing to write “Schemes” instead of more specific terms. For example, Scheme 1 is mentioned in two places (lines 308 and 315) and Scheme 2 in two places (lines 312 and 318). Therefore, I suggest changing all the “schemes” into proper Supplementary Figure numbers and Table numbers.

Figure-1. Mention the statistics used for Figure-1A in the legend (line 308).

In Figure 1B, in the y-axis, it is unclear what does the %  relative to.

Figure-2. Mention the statistics used for Figures in the legend (line 312).

Scheme 1(line 315) in the table, what does the number in parenthesis refers to ?. Also mention what n refers to (patients or samples ?)

Author Response

Dear Editor,

First and foremost, on behalf of all authors I would like to thank you and the Reviewers for the valuable comments and suggestions, which have significantly contributed to improving the overall quality of the manuscript. Each comment has been addressed below in point-by-point responses.

Best regards,

Dr Paul Billoir

INTRODUCTION: It would be useful to write a paragraph on the UFH, aPTT, anti-Xa activity, and Thrombin Generation Assay in the particular population of patients addressed in this study. At present, the introduction describes general “ICU cases”; however, the authors have used a selected set of patients. Therefore, it would be more meaningful to address the monitoring principles in this selected set of patients.

We understand reviewer suggestions. We have added in manuscript :

« aPTT is shortened in the presence of elevated factor VIII [4] or elevated fibrinogen [5]. Anti-Xa measurement is a direct UFH activity, with indirect quantification of antithrombin activity. Previous study have reported conflicting results for aPTT and anti-Xa measurement of heparin[6]. Several recommendations advocates if heparin resistance is suspected, anti–factor Xa can be used to measure the heparin level [7].”

RESULTS:

1).  Line 70. It seems multiple samples were taken from each patients, since 107 samples were obtained from 32 patients. Therefore, it would be useful to mention how many samples (minimum-maximum) were taken and at what time point they were taken and the rationale for this approach.

We understand reviewer comments.

We performed monitoring between same patient to evaluate thrombotic and bleeding development.

We have added in manuscript:

‘’Number of blood samples were between 2 to 4 for each patient, and between the 1st day and 4 day of hospitalization.’’ 

2). Line 78. More information is needed for the control group. Describe the demographics of the healthy volunteers; whether they match with the corresponding parameters of the ICU cases.

We understand reviewer comment.

Healthy volunteers were used to evaluate "in vitro" values and normal values of thrombin generation assay to compare with ICU population.

We have added in methods and results section:

‘’Healthy volunteers had no history of bleeding and thrombosis.’’

‘’Under 30 healthy volunteers, 14 were male with a median age of 38.2±11.8 years.’’

3). Line 86. It is mentioned that 49 samples were concordant; but how many patients does it correspond to? 

In general, it is confusing to use the number of samples, instead of the number of patients in the results and analysis. Since more than one sample was likely taken from each patient, the data may be skewed; the authors should consider including the number of patients within parenthesis, whenever the sample numbers are mentioned.

We understand reviewer comment.

In manuscript, we have already described this comment:

‘’Concerning hemorrhagic or thrombotic complications, 6 samples were excluded because the presence or absence of complications was not correctly identified at the time of sampling (analysis on 101 samples).’’

METHODS:

1). Line 229. Mention the method of spiking. What doses of UFH were used for spiking and how was the spiking done (volume and conditions).

We understand reviewer comments.

We have added in manuscript:

‘’ To evaluate, in vitro, in normal conditions, a pool of normal plasma from 30 healthy volunteers was spiked with different doses of UFH (Heparin Choay, 5000 UI/mL) to obtain several target ranges’’

2). Line 233. Explain how much blood per draw was collected.

We understand reviewer comment.

We have added in manuscript :

‘’Two citrated tubes were collected.’’

3). Line 238. Mention if storing PPP in -80 oC would affect the analysis of biomarkers.

We understand reviewer comment.

We have added in manuscript :

‘’ as recommended by the “Groupe Français d’études sur l’Hémostase et la Thrombose”, to perform aPTT, anti-Xa and TGA measurement (www.geht.org).’’

4). Line 272-274. Remove the Section 6. Patents

We performed reviewer suggestions.

5). Line 287. Mention who received the funding, the name of the funder, and the funding/project number.

This was a non-interventional research study. The study has been performed with standard of care, who not required funding.

It is confusing to write “Schemes” ;instead of more specific terms. For example, Scheme 1 is mentioned in two places (lines 308 and 315) and Scheme 2 in two places (lines 312 and 318). Therefore, I suggest changing all the “schemes” into proper Supplementary Figure numbers and Table numbers.

We understand reviewer comments.

We have kept the supplemental figures to simplify the reading

Figure-1. Mention the statistics used for Figure-1A in the legend (line 308).

We have added :

‘’Comparison was performed with Kruskall–Wallis ANOVA with Dunn’s multiple comparisons post-test.’’

In Figure 1B, in the y-axis, it is unclear what does the %  relative to.

% in supplemental figure 1B is unit for FVIII circulating concentration (normal range: 50-150%)

Figure-2. Mention the statistics used for Figures in the legend (line 312).

‘’Comparison was performed with Kruskall–Wallis ANOVA with Dunn’s multiple comparisons post-test or Mann-Whitney test.’’

Scheme 1(line 315) in the table, what does the number in parenthesis refers to ?. Also mention what n refers to (patients or samples ?)

N is the number of patients in each group.

For example, in first group, 27 had aPTT<T and anti-Xa<T. n corresponding patients with increased ETP.

We have added in figure legend:

‘’N: number of patient samples in each group. n: number of patient samples with increased ETP.’’